# Thermo- and pH-Responsive Gelatin/Polyphenolic Tannin/Graphene Oxide Hydrogels for Efficient Methylene Blue Delivery

**DOI:** 10.3390/molecules26154529

**Published:** 2021-07-27

**Authors:** Ariel C. de Oliveira, Paulo R. Souza, Bruno H. Vilsinski, Manuel E. G. Winkler, Marcos L. Bruschi, Eduardo Radovanovic, Edvani C. Muniz, Wilker Caetano, Artur J. M. Valente, Alessandro F. Martins

**Affiliations:** 1Group of Polymeric Materials and Composites, Department of Chemistry, State University of Maringá (UEM), Maringá 87020-900, PR, Brazil; pg54114@uem.br (A.C.d.O.); pg53548@uem.br (P.R.S.); bhvilsinski2@uem.br (B.H.V.); pg53546@uem.br (M.E.G.W.); ecmuniz@uem.br (E.C.M.); 2Chemistry Department, Institute of Exact Sciences, Federal University of Juiz de Fora, Juiz de Fora 36036-900, MG, Brazil; 3Laboratory of Research and Development of Drug Delivery Systems, Department of Pharmacy, State Univesity of Maringá (UEM), 5970 Colombo Avenue, Maringá 87020-900, PR, Brazil; mlbruschi@uem.br; 4Department of Chemistry, State University of Maringa, Maringá 87020-900, PR, Brazil; eradovanovic@uem.br; 5Department of Chemistry, Federal University of Piaui, Teresina 64049-550, PI, Brazil; 6Núcleo de Pesquisas em Sistemas Fotodinâmicos, Chemistry Department, State University of Maringá, Maringá 87020-900, PR, Brazil; wcaetano@uem.br; 7Department of Chemistry, CQC, University of Coimbra, 3004-535 Coimbra, Portugal; 8Laboratory of Materials, Macromolecules and Composites, Federal University of Technology—Paraná (UTFPR), Apucarana 86812-460, PR, Brazil

**Keywords:** physical hydrogels, drug delivery systems, condensed tannins

## Abstract

Gelatin (GE), amino-functionalized polyphenolic tannin derivative (TN), and graphene oxide (GO) were associated to yield thermo- and pH-responsive hydrogels for the first time. Durable hydrogel assemblies for drug delivery purposes were developed using the photosensitizer methylene blue (MB) as a drug model. The cooling GE/TN blends provide brittle physical assemblies. To overcome this disadvantage, different GO contents (between 0.31% and 1.02% *wt*/*wt*) were added to the GE/TN blend at 89.7/10.3 *wt*/*wt*. FTIR and RAMAN spectroscopy analyses characterized the materials, indicating GO presence in the hydrogels. Incorporation studies revealed a total MB (0.50 mg/mL) incorporation into the GE/TN-GO hydrogel matrices. Additionally, the proposed systems present a mechanical behavior similar to gel. The GO presence in the hydrogel matrices increased the elastic modulus from 516 to 1650 Pa. SEM revealed that hydrogels containing MB present higher porosity with interconnected pores. Dissolution and swelling degree studies revealed less stability of the GE/TN-GO-MB hydrogels in SGF medium (pH 1.2) than SIF (pH 6.8). The degradation increased in SIF with the GO content, making the polymeric matrices more hydrophilic. MB release studies revealed a process controlled by Fickian diffusion. Our results point out the pH-responsible behavior of mechanically reinforced GE/TN-GO-MB hydrogels for drug delivery systems purposes.

## 1. Introduction

Protein-based materials have been extensively used in biomedical applications due to their cytocompatibility and biodegradability. Thereby, gelatin-based hydrogels for drug delivery purposes are being explored. Physical hydrogel assemblies have received great attention because of their advantages compared to chemical hydrogels. These materials can be designed to avoid toxic chemistries (crosslinkers, organic solvents, and surfactants), which crosslink protein chains to support durability, using one-step strategies (in situ methods) [1]. These matrices comprise three-dimensional and hydrophilic structures (amine (–NH_2_), thiol (–SH), hydroxyl (–OH), and carboxyl (–COOH) groups) capable of absorbing biological fluids and swelling. These sites enable the incorporation of hydrophilic drugs within the hydrogel matrices by establishing electrostatic, H-bonding, and ion-dipole forces [2]. 

The gelatin (GE) protein has been extensively used to engineer hydrogels. It is obtained from collagen denaturation [3] and is composed mainly of proline, glycine, and 4-hydroxyproline in different contents [4]. The average molar mass varies from 30,000 to 65,000 g/mol depending on the natural source [4]. GE is an exciting material for biomedical purposes since it mimics well the connective tissue, has low cost, and is present in high availability [5]. However, this material presents low mechanical and thermal strength [3]. This limitation is a drawback for the direct GE application as drug delivery systems (DDS).

One viable alternative is associating GE with other polymers to increase the mechanical property of the final material. For example, hydrogels constituting GE and other polymers such as pectin, poly(acrylic acid), and polyacrylamide have been developed that aim for biomedical applications [2]. Previously, we proposed a physical hydrogel constituting GE/amino-functionalized tannin (TN) for biomedical applications involving wound dressing purposes [4]. TN is an inexpensive and abundant material easily extracted from the seeds of various plants, leaves, fruits, and barks [6]. TN is obtained from condensed tannin in the presence of formic acid and ammonium chloride. This polymer presents cationic moieties that can interact well with negatively-charged GE by hydrogen and ionic bonds [7]. Additionally, hydrophobic interactions can occur involving the polyphenolic moieties of TN and GE [8]. 

Our previous results revealed an assembly formation after the exposure of GE and TN on cooling the blends at 4 °C [4]. TN presence was essential to stabilize the GE helices and promote hydrogel formation. [4]. This stabilization was majority attributed to hydrophobic forces and hydrogen bonds between GE and TN. However, besides hydrogel formation, the related hydrogel presented aqueous instability [4]. 

To avoid the aqueous instability of the material, graphene oxide (GO) was added. GO is a highly oxidized form of the graphene molecule that uses its functional groups (epoxy (–O–), hydroxyl (–OH), and carboxylic (–COOH)) to conjugate with biomolecules and proteins [9]. These interactions give materials with recognized biocompatibility and improved physical and chemical properties, such as high fracture strength and high Young modulus [10]. The cooling of these blends provides physical hydrogels with aqueous stability. Here, for the first time, these hydrogels were designed as efficient drug delivery carriers. The phenothiazine compound Methylene blue (MB) was selected as a drug model because it is hydrophilic and presents high aqueous bioavailability. MB presents an aromatic and positively charged structure at biological pHs. It is also a Food and Drug Administration-approved drug that has been used as a photosensitizer in photodynamic therapy against several types of cancers and infections caused by different pathogens [11,12]. Its application is due to the excellent photophysical properties of MB, such as singlet oxygen generation in a biological medium, low price, reduced pain, and without several side effects [11]. Moreover, besides its water solubility, MB is quickly reduced to its non-active leuco-MB form in a biological medium [13]. In this way, to prevent this premature reduction, to increase the MB retention time in contact with the target cells, and promote the controlled delivery of MB, we propose the development of GE/TN-GO-MB hydrogels. 

Here, we intend to show that the physical hydrogels (GE/TN-GO) can incorporate hydrophilic drugs, such as MB. We propose new DDS for MB and related compounds based on physical, durable, and cytocompatible hydrogels.

## 2. Results and Discussion

### 2.1. Hydrogel Formation 

Figure 1 presents digital images of the GE/TN-GO-MB mixture at 50 °C and the obtained hydrogel after gelation at 4 °C for 1 h.

GE/TN-GO-MB hydrogels were obtained by mixing GE solutions and TN-GO-MB suspensions prepared in water (pH ≈ 5.5) at 50 °C. GE/TN-GO-MB mixtures are homogenous at 50 °C, providing physical hydrogels by gelation at 4 °C (Figure 1). Aqueous GE solutions contain protein chains in helical conformation due to the effective establishment of H-bonds between them [14]. Thus, the gelation of GE mixtures can occur by protein stabilization. The TN (Zeta potential equal to ≈−15 mV at pH 5.5) can stabilize the GE (type B) chains mainly by intermolecular interactions. Coulomb forces are established between protonated groups and anionic carboxylate ions in TN and GE. The GE isoelectric point is between 4.8–5.1. Therefore, protonated GE moieties can electrostatically interact with hydrolyzed tannins at pH 5.5. The GE and TN chains also present hydrophilic sites, including –OH, which can interact with H-bonds [4]. 

A pure GE/TN assembly was created from a GE solution at 4% wt/vol (4-GE/TN) and dialyzed TN at 4.6 × 10^−3^ mg/mL at a 50% *wt*/*wt*. The GE/TN weight ratio presented high stability in the water against disintegration and dissolution [4]. Therefore, the 4-GE/TN weight ratio previously reported was selected to create GE/TN-GO-MB hydrogels. However, the pure 4-GE/TN hydrogels showed brittle structures with poor mechanical properties to be used as DDSs. 

To overcome this disadvantage and improve the hydrogel mechanical properties GO (0.31, 0.51, 0.72, and 1.02% *wt/wt*) was associated with the hydrogels. GO has been explored for enhancing the mechanical properties of materials, by being applied as mechanical reinforcement for hydrogel matrices in biomedical applications [15]. GO contains peripheral carboxylic sites (–COOH) that provide stability and negative charge dependency on pH (–COO^−^). The epoxy (–O–) and hydroxyl (–OH) groups at the basal plane allow for weak interactions and H-bonds. In addition, one must take into account the π electrons present due to unmodified areas of graphene, in which they are hydrophobic and capable of interacting by π-π interactions [15]. These GO interactions occur with TN and GE.

As shown in Figure 1, the blue coloration indicates the presence of MB in the hydrogels. To add MB in the hydrogels, an in situ strategy was used. This strategy prolongs the medication’s action, improves patient compliance, and reduces medication administration frequency compared to the conventional medication delivery system [16]. Cationic MB interacts with hydrogels by coulombic, ion-dipole, H-bonds, and hydrophobic forces [13]. These interactions also occur more precisely between the MB imines groups and the GE, TN, and GO networks’ charged groups (Scheme 1) [17]. 

### 2.2. Characterization 

The dynamic rheology was used to evaluate the GE/TN-GO-MB mixture properties by measuring storage modulus (G’) and loss modulus (G’’). These properties characterize the viscoelastic behavior and the temperature for the sol–gel transition (T_sol-gel_) of the systems [18].

Figure 2 shows curves of G’ and G’’ as a function of the temperature. The cross-over point between G’ and G’’ determines the temperature that indicates the liquid–solid phase transition. The elastic behavior of GE/TN-GO-MB mixtures prevails at temperatures up to 60 °C because the values of G’ are greater than G’’ (Figure 2). As shown, the G’ and G” curves do not provide a crossing point in a temperature range between 5 and 60 °C for all samples, indicating that the T_sol-gel_ occurs above 60 °C. In fact, there is a cross-over trend in the G’ and G” curves close to 60 °C. However, the GE chains degrade above 60 °C, avoiding measures of G’ and G’’.

Figure 3 shows G’ and G” curves as a function of the oscillatory frequency for the assemblies at 37 °C. The hydrogel’s structural consistency can be assessed by analyzing G’ and G”. The gel state prevails when G’ is higher than G”. As G’ is lower than G’’, the energy used to deform the material is dissipated, indicating a liquid behavior [19,20]. The elastic response (G’) predominates over the viscous flow (G”) in the entire frequency range analyzed (Figure 3). Therefore, the GE/TN-GO-MB systems have mechanical spectra similar to gels. Our findings agree with other literature findings, confirming that the gel state prevails upon the liquids state at a high GE content [4].

The successful application of hydrogels in controlled drug release is related to their mechanical characteristics. In case of failure or reduced mechanical strength, the hydrogel’s integrity under certain conditions can be affected [21]. Thereby, Young’s or the elastic modulus of the wet hydrogels was measured. Table 1 shows the effect of the GO content (% *wt*/*wt*) on hydrogel mechanical properties. The elastic modulus is between 516 ± 1.62 and 1650 ± 33.60 Pa. The elastic modulus significantly increases as the GO content is raised in the material structure. The GE/TN-MB elastic modulus without GO is 516 Pa, whereas the 4-GE/TN-GO-MB is 1650 PA (Table 1). These results agree with the results found by [22]; i.e., increasing the GO content (0–6% *vol*/*vol*), shows higher values in Young’s modulus in the GO/epoxy composite [22]. 

Figure 4 presents the FTIR spectra of the GE, dialyzed TN, MB, and hydrogels (1-GE/TN-GO-MB, 2-GE/TN-GO-MB, 3-GE/TN-GO-MB, and 4-GE/TN-GO-MB). The GE FTIR spectrum (Figure 4A(i)) presents characteristic bands at 1634 cm^−1^ assigned to the C=O stretching of amide (I) and carboxylate ions at 1536 cm^−1^ ascribed to the angular stretching of N–H and amide (II) bonds. The band at 1242 cm^−1^ is related to the stretching of amide (III), whereas the signal at 3238 cm^−1^ is ascribed to the –OH stretching [4]. These bands confirm the GE structure.

The TN FTIR spectrum (Figure 4A(ii)) presented a band at 1717 cm^−1^ assigned to the C=O stretching of carboxylic acids due to the presence of hydrolyzed tannins in the TN structure [23]. The bands at 1623 and 1092 cm^−1^ are attributed to the C=C and C−O stretching, respectively, found on condensed tannins’ phenolic groups. The band at 771 cm^−1^ indicates the angular stretching of C=C−H bonds on aromatic rings [4]. These signals show that the TN comprises hydrolyzed and polyphenolic tannins.

The MB FTIR spectrum (Figure 4A(iii)) shows vibrational modes at 1591 and 1387 cm^−1^ assigned to the vibrations of the aromatic rings; 1326 cm^−1^ is ascribed to the C−N stretching; 1491 indicates vibrations of the aromatic ring; 1245, 1137, and 1032 cm^−1^ indicate the C−H (in the plane) bending vibrations; 879, 822, 665 cm^−1^ assigned to the C−H (out of plane) bending vibrations [24,25].

The hydrogels’ FTIR spectrum (Figure 4B) presents characteristic bands assigned to the GE, TN, and MB. These confirm that GE, TN, and MB are assembled in the hydrogels. The broadband between 3436 cm^−1^ in the assembly FTIR spectra indicates effective H-bonds between the GE and TN chains. The band at 805 cm^−1^ is assigned to the C=C–H stretching found on aromatic rings in condensed tannins [4]. The bands at 1689 and 1082 cm^−1^ are attributed to the C=C and C−O stretching, confirming TN’s presence in the hydrogels. The signals at 1237, 1532, 1632, and 1689 cm^−1^ indicate stretching of amide (III), angular stretching of N–H and amide (II), and C=O (amide (I) and carboxylate ions), respectively. These bands are attributed to GE’s presence in the hydrogels. The vibrational modes at 1387 and 585 cm^−1^ in the hydrogel FTIR spectra are related to MB. The bands are attributed to the aromatic rings’ vibrations and the C–H out-of-plane vibrations. The FTIR analysis confirms that the hydrogels are comprised of GE, TN, and MB.

Raman spectroscopy (Figure 5) was used to characterize carbon materials because the spectral shape exhibits a wide variety corresponding to the carbon forms, revealing structural information [26]. Through this technique, the GO and its presence in the hydrogels were analyzed according to Figure 5. The Raman spectrum of GO (blue line) shows a 2D-band at 2682 cm^−1^, G-band at 1568 cm^−1^, and D-band at 1345 cm^−1^. The G-band is associated with graphitic carbons; 2D-bands and D-band are related to the structural defects or partially disordered graphitic domains [27,28]. 

In the hydrogels’ spectra, the G bands at 1568 cm^−1^ and 2D bands at 2682 cm^−1^ are observed. The D-band is also subtly noted at 1345 cm^−1^ in the 3-GE/TN-GO-MB and 4-GE/TN-GO-MB hydrogels. This event is not observed in the other hydrogels. The 3-GE/TN-GO-MB and 4-GE/TN-GO-MB hydrogels consist of 2.23 and 3.05% *wt*/*wt* of GO in the matrix, respectively. Therefore, these bands confirm the presence of GO in hydrogels. 

Figure 6 shows SEM images of the lyophilized hydrogel cross-sections. The hydrogels incorporated with MB present high porosity, between 64.54 and 77.26% (Appendix A) with no phase distinction and interconnected pore networks. The porosity facilitates water diffusion toward the three-dimensional matrices, favoring the release of solutes [29]. Porous and three-dimensional matrices are suitable DDSs [30].

### 2.3. Disintegration/Dissolution In-Vitro and Water Uptake

Figure 7 shows the disintegration results in SIF (pH 6.8) and SGF (pH 1.2) at 37 °C. The hydrogels exhibit lower stability in SGF than in SIF. At pH 1.2, the GE chains degrade, disintegrating the hydrogels [4]. Therefore, in SGF, all the physical assemblies present complete disintegration after 2 days (Figure 7A). 

The hydrogels are completed disintegrated in SIF after 4 days. Overall, the disintegration percentage increases as the GO content raises in the hydrogel (*p* < 0.05). The disintegration increases from 39.89 ± 2.43 to 57.93 ± 3.29% after 1 day for 1-GE/TN-GO-MB and 4-GE/TN-GO-MB, respectively. The SIF (pH 6.8) promotes ionized –COO− upon GO (isoelectric point < 2.0), making the hydrogels hydrophilic, increasing the disintegration/dissolution percentage [31].

Figure 8 shows the swelling degree results in SIF (pH 6.8) and SGF (pH 1.2) at 37 °C. The hydrogels swell more in SGF (swelling degree between 2346 ± 231% and 3107 ± 183%) than in SIF (swelling degree between 721 ± 84% and 1238 ± 166%) after 1 day. SGF provides an excess of H_3_O^+^ and the MB is protonated, imparting MB_2_^+^ species [32]. As a result, there is repulsion between the hydrogel structures and H_3_O^+^ ions, consequently, with more significant swelling. 

The swelling degree in SIF (Figure 8B) is 1238 ± 166%, 1139 ± 50%, and 1191 ± 60% for the hydrogels 1-GE/TN-GO-MB, 2-GE/TN-GO-MB, and 3-GE/TN-GO-MB, respectively. On the other hand, the hydrogel 4-GE/TN-GO-MB behaved differently after 1 day. This hydrogel has the lowest swelling degree (721 ± 84%) if compared with others, because it disintegrates 57.93 ± 3.29% in this medium, as previously mentioned. For instance, this greater degradation (57.93 ± 3.29%) compared to other hydrogels (between 39.89 ± 2.43 and 47.18 ± 2.44 to 1-GE/TN-GO-MB, 2-GE/TN-GO-MB, and 3-GE/TN-GO-MB, respectively), decreases the swelling capacity of the material, due to its mass loss. Finally, this pH-sensitive swelling behavior should favor the controlled release of bioactive compounds [33].

### 2.4. Methylene

Figure 9 shows MB releases in vitro. These releases were evaluated with the dried hydrogels obtained after lyophilization in SGF (pH 1.2) and SIF (pH 6.8), simulating the stomach and intestine pH conditions, respectively. The hydrogels 3-GE/TN-GO-MB and 4-GE/TN-GO-MB were selected to evaluate the MB release because of their high elastic modulus.

In SGF, dried hydrogels released low MB contents, achieving between 13.82% (17.96 × 10^−3^ mg/mL) and 13.45% (±1.22%) (16.14 × 10^−3^ mg/mL) for 3-GE/TN-GO-MB and 4-GE/TN-GO-MB hydrogels, respectively. The SIF released 7.33% (±0.74) (9.53 × 10^−3^ mg/mL) and 9.54% (±0.13) (11.44 × 10^−3^ mg/mL) for the 3-GE/TN-GO-MB and 4-GE/TN-GO-MB hydrogels, respectively. 

Of note is that the maximum released MB content was achieved after 24 h in SGF and 72 h in SIF. The hydrogels disintegrated in both SGF and SIF after 24 and 72 h, releasing 100% of the incorporated MB (Appendix A). The faster degradation of GE/TN-GO-MB hydrogel in SGF agrees with the swelling degree/disintegration studies. These studies showed a higher swelling and erosion rate in the acidic medium (as described previously in Section 2.3). This result is similar to previous studies performed in our group involving the solubilization and controlled release of MB from gum Arabic hydrogels [13]. 

Gastrointestinal transit time is an important factor for dosage forms and medications. Gastric transit can vary from 0 to 2 h on an empty stomach and can be prolonged for up to 6 h when fed into the small intestine for around 3–4 h [34]. Taking these results into account, it can be inferred that the drug release time is less than the hydrogel degradation time in the body (24 and 72 h); i.e., this degradation does not preclude the proposal of the material as DDS. It should be noticed that all materials presented here are biocompatible. According to Dinescu et al., up to 3% *wt*/*wt* (0.075 g) of GO used showed biocompatibility in healthy cells (mouse preosteoblasts), enabling the use of this material as DDSs [35]. The GE/TN-GO hydrogels displayed pH-responsive behavior, controlling the MB release. The MB was selected as a cationic hydrophilic drug model in our studies. Therefore, drugs with similar polarities compared to MB can be released from the GE/TN-GO matrices.

### 2.5. Transport Mechanism of MB from the Hydrogels 

The release process can occur as a diffusional transport process and/or with a partition phenomenon between the solvent and solid hydrogel phases. Several models have been applied to elucidate the release mechanisms of solutes from hydrogel matrices, including zero-order, first-order, second-order, Higuchi Model, Hixson–Crowell cubic root equation, and Korsmeyer–Peppas kinetic models. 

The Korsmeyer–Peppas model fits best with the MB release curve in both media for the 3-GE/TN-GO-MB and 4-GE/TN-GO-MB hydrogels. Table 2 shows the parameters *n* and *k* presented obtained from the release curves in Figure 9. The fitting by the other models presents low correlation coefficients (*R*^2^ lower than 0.70). Figure 9 shows the release kinetics for hydrogels in SGF and SIF. The Korsmeyer–Peppas model proposes a semi-empirical model (Equation (1)) that describes the transport of solutes from a flexible matrix.
(1)$$\ln\frac{C_{t}}{C_{\infty }}=\ln k+nt$$

C_t_ and C∞ are the cumulative concentrations of MB at time *t* and infinite, respectively. The *k* and *n* are the fit parameters, where n is the diffusional exponent and represents the release mechanism. The values of *n* depend on the hydrogel’s geometric shape [36].

For the hydrogels’ release in a cylindrical shape, the release mechanism depends on the values of *n*. When *n* is around ≤0.45, the drug release mechanism is controlled by Fickian diffusion. This result indicates that the matrix’s drug diffusion is a determining step in the release process [36,37]. For *n* > 0.89, the mechanism is considered a Super Case II. In this case, the swelling process or macromolecular relaxation increases the hydrogel chains’ mobility in contact with the water [36,37]. When *n* is between 0.45 and 0.89, an anomalous or non-Fickian transport is observed, in which the release of solutes depends on the simultaneous diffusion and hydrogel matrix relaxation [37,38]. 

The *n* value for all hydrogels was less than 0.28 in both mediums. According to works reported elsewhere [36,37], this behavior is controlled by the Fickian diffusion. This is because the diffusion rate is slower than the time required for hydrogel chain relaxation [36,37]. However, it is necessary to mention that the Korsmeyer–Peppas model is used to evaluate the first 60% of the total amount of MB [39]. Therefore, the entire MB release should be controlled by a more complex behavior involving several factors, such as polymer-drug interaction, matrix erosion, and others [40]. 

## 3. Materials and Methods

### 3.1. Materials

Gelatin (GE, type B), extracted from bovine bones, was donated by Rousselot Gelatinas SA (Amparo, São Paulo, Brazil). The amino-functionalized polyphenolic tannin (TN; 4.6 × 10^−3^ mg/mL) derivative, commercially named tanfloc SG, was graciously donated by Tanac SA (Montenegro, Rio Grande do Sul, Brazil). Graphene oxide (GO) was prepared by the chemical oxidation of graphite powder according to the modified Hummers method [41]. Methylene blue (MB, 373.90 g/mol, 85%) and the dialysis bag (cut-off 12 kDa) were purchased from Sigma-Aldrich (São Paulo, Brazil).

### 3.2. The Amino-Functionalized Polyphenolic Tannin (TN) Purification

The dialysis process removes impurities (calcium, potassium, chloride ions, and formic acid) from the TN structure [4]. For this, a TN solution (5.0% wt/vol) was prepared in distilled water (pH ≈ 5.5), previously filtered to remove wood particles, and dialyzed against distilled water for five days. The water exchange was performed twice for each day. After dialysis, the as-obtained TN solution was immediately used to obtain GE/TN-GO-MB mixtures. The TN concentration remaining in the dialysis bag after five days was 4.6 × 10^−3^ mg/mL. The dialyzed TN solution was frozen and lyophilized. The TN concentration was determined following an experimental procedure reported elsewhere [4].

### 3.3. Hydrogel Preparation

Hydrogels based on GE/TN were developed following a reported experimental procedure [4] with alterations. GO suspension as well as aqueous MB and GE solutions were prepared individually. The GO was dispersed in 10 mL of deionized water performing a GO suspension (0.10 mg/mL) at room temperature, ranging in 0.31, 0.51, 0.72, and 1.02% *wt/wt*. The MB solution was also prepared in deionized water (10 mL) at 1.58 × 10^−3^ mol/L (0.50 mg/mL) at room temperature. Then, the GO suspension (10 mL) was mixed with the dialyzed TN solution (50 mL) under shaking (400 rpm) at 50 °C for 10 min, obtaining the TN-GO mixture. Next, the prepared MB solution (10 mL) was mixed with the TN-GO mixture under shaking (400 rpm) at 60 °C for 10 min, obtaining the TN-GO-MB mixture. Finally, the TN-GO-MB mixture final volume ranged from 72.1 and 77.0 mL, depending on the % *wt/wt* of GO. The aqueous GE solution (4.0% wt/vol) at 50 °C was mixed with the TN-GO-MB mixture (1.5 mL) at 50 °C. The TN-GO-MB suspension was slowly dropped in the GE solution (1.5 mL, 50 °C), obtaining a GE/TN-GO-MB mixture (6.0 mL) at a 50:50 GE/TN-GO-MB volume ratio. Table 3 presents the experimental conditions used to create physical assemblies based on GE/TN-GO-MB. After 5 min at 50 °C, the GE/TN-GO-MB mixture was cooled (4 °C) for 1 h, supporting the physical hydrogel assemblies containing different GO contents (Table 3). The materials were frozen and lyophilized for 48 h for further analysis. The assemblies were denoted as x-GE/TN-GO-MB, where x is the relative % *wt/wt* of the GO solution used to create the mixtures (Table 3).

### 3.4. Characterization 

The purified TN, GE, GO, MB, and GE/TN-GO-MB hydrogels were characterized by Fourier-transformed infrared spectroscopy (FTIR). FTIR spectra were recorded using a Fourier-transformed infrared spectrophotometer (Thermo Fisher Scientific Inc. (Waltham, MA, USA), operating between 400 and 4000 cm^−1^ with a resolution of 4 cm^−1^ and accumulation of 64 scans. Raman spectra were measured using a Witec Alpha 300 Raman spectrometer (USA) with 1800 lines/mm using a 532 nm laser. 

The rheological properties of the GE/TN-GO-MB mixtures were measured using a controlled stress rheometer (MARS II Haake^®^) equipped with a 35 mm diameter cone plate separated by a fixed distance of 0.052 mm. Samples were carefully placed on the lower plate to ensure the minimal shear of the sample. The linear viscoelastic range (LVR) was determined for each formulation from 5 to 60 °C. The LVR is obtained when the strain and deformation are proportional between them, and G’ moduli remain constant. The temperature sweep was performed at a range from 5 to 60 °C with a 10 °C/min heating rate, followed by a constant strain at 1.0 Hz. The dynamic rheological properties, storage modulus (G’), and loss modulus (G’’) were determined by using the RheoWin 4.10.0000 (Haake^®^, Ober-Moerlen, Germany) software. In each case, at least three replicates’ viscoelastic properties were determined (*n* = 3). G’ and G’’ curves as a function of the temperature were obtained. The gelation temperatures (Tsol-gel) occur when the G’ and G’’ have the same value. In the oscillation mode, the analysis was carried out at 37.0 ± 0.1 °C, using the same rheometer and geometry previously reported. After LVR determination, the frequency sweep was performed from 0.1 to 10.0 Hz under constant stress [4]. The wet hydrogels’ elastic modulus was measured using a universal test machine (Ametek LR10K PLUS, West Sussex, UK), following a methodology reported elsewhere [21]. 

The hydrogel morphology was investigated by scanning electron microscopy (SEM) using an FEI-QUANTA 250 (Czech Republic) microscope at 15 kV of accelerating voltage. The samples were dispersed in double-sided tape coated with a thin gold layer (≈50 nm) for SEM images. Differential scanning calorimetry (DSC) analysis was performed in a calorimeter Shimadzu DSC 60 Plus (Kyoto, Japan) at 10 °C/min between 20 and 300 °C under 50 mL/min of Argon purge.

The hydrogel porosity was determined following an experimental protocol reported elsewhere [42]. The hydrogels (swelling) were weighed (m_1_) and oven-dried under vacuum (250 mmHg) at 37 °C. The dried materials with constant weights (m_2_) were obtained by removing the free water. The hydrogel porosity (%) was calculated by Equation (2).
(2)$$\mathrm{Porosity}=\left\{\frac{\frac{\left(m_{1}-m_{2}\right)}{\mathrm{Pw}}}{V}\right\}\times 100\left(\%\right)$$
where m_1_ and m_2_ are the wet and dry hydrogel weights. Pw is the specific mass of water at 37 °C and V is the wet hydrogel volume.

### 3.5. Hydrogel Disintegration/Dissolution

The initial dry weight (W_d–I_) of the hydrogel was measured after the lyophilization. Then, dried samples (≈0.10 g) were added to the simulated gastric fluid (SGF (pH 1.2): 2.0 g NaCl and 7.0 mL of the concentrated aqueous HCl solution (37% wt/vol) in 1000 mL of water) and phosphate-buffer (SIF (pH 6.8): 6.80 g of KH_2_PO_4_ and 77 mL aqueous NaOH 0.20 mol/L in 1000 mL of water) and incubated at 37 °C under shaking (100 rpm) [43]. At desired time intervals (after 1, 2, 3, 4, and 5 days), the hydrogels were removed from the solutions, oven-dried at 35 °C for 24 h, and the final dry weights (W_d–F_) were determined. The disintegration/dissolution (%) was assessed through Equation (3) (*n* = 3).
(3)$$\frac{\mathrm{Disintegration}}{\mathrm{dissolution}}\left(\%\right)=\;\frac{W_{d-I}-W_{d-F}}{W_{d-I}}\times 100$$

### 3.6. Swelling Degree

The lyophilized hydrogels’ swelling degree (W_d_) was obtained at 37 °C in SIF (pH 6.8) and SGF (pH 1.2) after 1, 2, 3, 4, and 5 days of exposure. The swelling degree (%) was determined by Equation (4) [44].
(4)$$\mathrm{Swelling}\;\mathrm{degree}\;\left(\%\right)=\;\frac{W_{s}-W_{d}}{W_{d}}\times 100\%$$
where W_s_ and W_d_ are the weights of swollen and dried hydrogels, respectively. The assays were performed in duplicate (*n* = 2).

### 3.7. Methylene Blue Release In Vitro

The MB release assays were performed according to the methodologies already reported with alterations [13] in SGF and SIF for 7 days. Dried hydrogels (20 mg) were added to sealed flasks containing SGF or SIF (20 mL) without shaking at 37 °C. At desired time intervals, aliquots (2.0 mL) were removed from the flasks and centrifuged (5.0 min at 4000 rpm). The released MB content as a function of time was spectrophotometrically measured by analyzing the MB absorbance at 664 nm, taking into account the molar absorptivity coefficient ɛ for MB in SIF (63,000 L/mol cm) and SGF (37,500 L/mol cm) [13]. A control experiment was performed by adding 20 mg MB for 7 days at 37 °C in both SIF and SGF solutions.

### 3.8. Statistical Analysis

The results were statistically analyzed using ANOVA and Tukey tests at a 5% significance level (GraphPad Prism 6.0).

## 4. Conclusions

Physical and thermosensitive hydrogel assemblies based on GE, TN, GO, and MB were successfully prepared. Rheological assays showed that GE/TN-GO-MB assemblies display gel behavior with gelation above 60 °C. The FTIR analysis indicated that GE and TN are essential to forming hydrogel assemblies. Raman spectroscopy showed the presence of GO as being essential for the results of the mechanical property. We created porous and pH-responsive hydrogels loaded with MB. We showed that the MB comprises the hydrogel structures. MB release studies were carried out in different environments (SGF and SIF), using the hydrogels as DDSs. The hydrogels proposed were effective in promoting the pH-responsible behavior for MB release.

Finally, we present for the first time hydrogels based on natural sources (GE and TN) containing GO for MB release. The hydrogels were obtained using only deionized water as a solvent. The approach avoids the use of chemical crosslinking agents, which often crosslink GE-based materials. The results support the continuation of the studies involving the use of this platform for drug delivery purposes.

## Data Availability

Not applicable.

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
