# Peer review of "Thermo- and pH-Responsive Gelatin/Polyphenolic Tannin/Graphene Oxide Hydrogels for Efficient Methylene Blue Delivery"

_molecules, 2021, doi:10.3390/molecules26154529_

Round 1
Reviewer 1 Report
Authors present new material in the form of physical blend for application in photodynamic therapy based on gelatin and tannin with some addition of graphene oxide as structuring factor. Material was carefully characterized by rheological methods and spectroscopy. The range of stability was in accordance with the scope of medical application. New blend successfully released MB.
Article is well written and addressess all points necessary to evaluate its applicability. However it would be nice to have information if there is some excess of GO still present in the blend and if yes, than what might be impact of this additive on medical safety.
Author Response
We really appreciate the comment raised by the Reviewer. The following phrase has been added in Section 2.4: "It should be noticed that all materials presented here are biocompatible. According to Dinescu et al., up to 3% wt/wt (0.075 g) of GO used showed biocompatibility in healthy cells (mouse preosteoblasts) , enabling the use of this material as DDSs"
Reviewer 2 Report
EVALUATION
Paper Title: “Thermo- and pH-Responsive Gelatin/Polyphenolic Tannin/Graphene Oxide Hydrogels for Efficient Methylene Blue Delivery”
”
Authors: Ariel C. de Oliveira, Paulo R. Souza, Bruno H. Vilsinski, Manuel E. G. Winkler, Marcos L. Bruschi, Eduardo Radovanovic, Edvani C. Muniz, Wilker Caetano, Artur J. M. Valente, Alessandro F. Martins
Comments:
This manuscript describes an interesting study concerning a novel thermo--and pH-responsive hydrogel made of gelatin, functionalized tannin and graphene oxide. The authors employed a multi-technique approach (FTIR and Raman spectroscopy, DSC, SEM, Rheological measurements and stability and release tests) to investigate the structural and functional features of this system. The methylene blue compound was used as a model drug for in vitro drug delivery studies. The manuscript is well written and the results are convincing.
Below I present some general suggestions/comments concerning the work:
- In the 2.2 Characterization section (line 260) the authors wrote “...as previously mention. I.e., this great degradation..” In this case, I believe it would be better to write: For instance, this great degradation..
- In the section 2.4 Release of Methylene Blue (line 270-271) the authors wrote “simulating the intestine and the stomach pH conditions.” It would be better to write: simulating the stomach and the intestine pH conditions.
Author Response
We thank the Reviewer for his/her positive comments on the ms. The modifications suggested by the Reviewer have been accepted and the ms has been modified accordingly.